# Optimization of Maduramicin Ammonium-Loaded Nanostructured Lipid Carriers Using Box–Behnken Design for Enhanced Anticoccidial Effect against *Eimeria tenella* in Broiler Chickens

**DOI:** 10.3390/pharmaceutics14071330

**Published:** 2022-06-23

**Authors:** Yan Zhang, Runan Zuo, Xinhao Song, Jiahao Gong, Junqi Wang, Mengjuan Lin, Fengzhu Yang, Xingxing Cheng, Xiuge Gao, Lin Peng, Hui Ji, Xia Chen, Shanxiang Jiang, Dawei Guo

**Affiliations:** 1Engineering Center of Innovative Veterinary Drugs, Center for Veterinary Drug Research and Evaluation, College of Veterinary Medicine, Nanjing Agricultural University, 1 Weigang, Nanjing 210095, China; mszy788@163.com (Y.Z.); zrn910314@163.com (R.Z.); sxh19940207@163.com (X.S.); gongjh365@163.com (J.G.); 2021207003@stu.njau.edu.cn (J.W.); lmj19980112@163.com (M.L.); yangfengzhu1001@163.com (F.Y.); 15551753051@163.com (X.C.); vetgao@njau.edu.cn (X.G.); pl18951630156@163.com (L.P.); jihui1019@163.com (H.J.); jiangshanxiang@163.com (S.J.); 2College of Animal Science and Technolog, Nanjing Agricultural University, 1 Weigang, Nanjing 210095, China; chenxia0123@njau.edu.cn

**Keywords:** maduramicin ammonium, nanostructured lipid carriers, Box–Behnken design, anticoccidial effect

## Abstract

Maduramicin ammonium (MAD) is one of the most frequently used anticoccidial agents in broiler chickens. However, the high toxicity and low solubility of MAD limit its clinical application. In this study, MAD-loaded nanostructured lipid carriers (MAD–NLCs) were prepared to overcome the defects of MAD by using highly soluble nanostructured lipid carriers (NLCs). The formulation was optimized via a three-level, three-factor Box–Behnken response surface method. Then, the optimal MAD–NLCs were evaluated according to their hydrodynamic diameter (HD), zeta potential (ZP), crystal structure, encapsulation efficiency (EE), drug loading (DL), in vitro release, and anticoccidial effect. The optimal MAD–NLCs had an HD of 153.6 ± 3.044 nm and a ZP of −41.4 ± 1.10 mV. The X-ray diffraction and Fourier-transform infrared spectroscopy results indicated that the MAD was encapsulated in the NLCs in an amorphous state. The EE and DL were 90.49 ± 1.05% and 2.34 ± 0.04%, respectively, which indicated that the MAD was efficiently encapsulated in the NLCs. In the in vitro study, the MAD–NLCs demonstrated a slow and sustained drug release behavior. Notably, MAD–NLCs had an excellent anticoccidial effect against *Eimeria tenella* in broiler chickens. In summary, MAD–NLCs have huge potential to form a new preparation administered via drinking water with a powerful anticoccidial effect.

## 1. Introduction

Coccidiosis, caused by *Eimeria,* is a common intestinal disease in poultry that induces a high mortality, reduces growth, and generates an annual loss of up to USD 3 billion in the global poultry industry [1]. *Eimeria tenella* is one of the most pathogenic and harmful organisms inhabiting the cecum. Chickens infected with *Eimeria tenella* exhibit depression, growth inhibition, and bloody stools, which eventually lead to death; the mortality rate of *Eimeria tenella* is over 80% [2]. Chemotherapy is the most effective treatment for coccidiosis. Maduramicin ammonium (MAD), an excellent and typical polyether ionophore antibiotic, is extensively used as an anticoccidial in clinical applications and has a stronger anticoccidial effect than other drugs [3].

MAD is the ammonium salt form of maduramicin extracted from the fermentation product of *Actinamadia yumaese* [4]. MAD is an important polyether ionophore antibiotic due to its high potency, broad anticoccidial spectrum, low cost, and durability against drug resistance. However, MAD has a highly toxic and a narrow safety range—the toxic dose (7.5 ppm) is very close to the clinically recommended dose (5 ppm) [5]. MAD does not easily dissolve in water, and therefore, it is usually mixed with feed, which often causes chicken poisoning in veterinary clinical use due to uneven mixing [6]. Furthermore, coccidiosis diminishes chickens’ appetites, sometimes entirely, but does not affect their thirst [7]. Thus, oral administration via drinking water is a more convenient way to deliver enough drugs to infected chickens [8]. Therefore, it is of urgent need to develop a new drug delivery system for MAD to improve its solubility and facilitate its administration via drinking water.

Nanostructured lipid carriers (NLCs) are a novel lipid-based nano-delivery system prepared by mixing solid and liquid lipids at a certain temperature [9]. NLCs prevent encapsulated drugs from being squeezed out during storage by using the added liquid lipids to form lipid skeletons containing nano-compartments. They improve upon earlier drug loading systems, such as unstable nanoemulsions and solid lipid nanoparticles (SLNs) with poor drug loading [10,11]. The addition of liquid lipids breaks the perfectly ordered crystalline structure of the solid lipids and leaves enough space to accommodate the drug molecules, further improving the drug loading capacity and stability of the NLCs [12]. Researchers have explored the ability of NLCs to accommodate large amounts of bioactive components [13], prevent their premature decomposition [14], and enhance their oral bioavailability [15]. In addition, the various surfactants employed in NLC formulations could overcome P-gp efflux and enhance the intestinal permeability by disrupting membranes [16]. More importantly, the highly favorable pharmacokinetic parameters of drug-loaded NLCs [17,18,19,20] make NLCs a promising tool for controlling the release of encapsulated drugs and enhancing their therapeutic efficacy. The preparation methods for NLCs include solvent dispersion, microemulsion, high-shear ultrasound, and high-pressure homogenization [21]. The high-shear ultrasound method has low cost, no organic solvent addition, and excellent stability properties of NLCs; however, its particle distribution is too broad and its yield too low, so it is often used for small-scale preparation and the screening of NLCs in laboratories [22]. The high-pressure homogenization method is widely used due to its simplicity, low cost, and capacity for large-scale production; the prepared NLCs are small and uniformly dispersed, with good stability [23]. Taken together, NLCs are a potent oral delivery carrier for insoluble drugs with promising marketability [24].

In this study, we developed the novel maduramicin ammonium-loaded nanostructured lipid carriers (MAD–NLCs) for the first time. The preparation process was optimized using the high-shear ultrasound method and the Box–Behnken response surface technique. The optimal MAD–NLCs were fabricated using high-pressure homogenization, and then, physiochemical characterization and a pharmaceutical analysis were conducted. Finally, the therapeutic effect of the MAD–NLCs on chickens infected with *Eimeria tenella* was investigated via drinking water administration for providing the basis for clinical treatment.

## 2. Materials and Methods

### 2.1. Materials

Maduramicin ammonium (MAD, 91%, *w*/*w*) was provided by the Zhejiang Huineng Biotechnology Company (Haining, China). The MAD reference substance (92.3%, *w*/*w*) was supplied by the China Veterinary Drug Inspection Institute (Beijing, China). Stearic acid, oleic acid, and Tween 80 were obtained from Aladdin (Shanghai, China). Poloxamer 188 was supplied by the Shanghai Yunhong Chemical Preparation and Accessories Technology Corporation (Shanghai, China). Acetonitrile and methanol were bought from Tedia (Fairfield, OH, USA).

### 2.2. Preparation of MAD–NLCs

#### 2.2.1. Screening of Lipids

MAD was dissolved in various solid lipids (lauric, stearic, and palmitic acids) at a temperature 15 °C higher than their melting points, and the weight of MAD was recorded until the drug could no longer dissolve. Meanwhile, the mixture was photographed under an optical microscope (40×) to determine whether the drug precipitated after cooling. Further, MAD was dissolved in various liquid lipids (castor oil, squalene, and oleic acid) at 25 °C. Herein, the solubility of MAD in each of the solid and liquid lipids was evaluated. Additionally, in order to determine their proportions for the follow-up experiments, the solid and liquid lipids were mixed in ratios of 1:9, 5:5, and 9:1, and the solubility of MAD was further evaluated.

#### 2.2.2. Screening of Emulsifiers to Mixed Lipids

Tween 20 (T20), Tween 60 (T60), Tween 80 (T80), Span 80 (S80), Poloxamer 188 (P188), and Poloxamer 407 (P407) were selected to investigate their emulsification effect on the solid–liquid mixed lipids to determine the main emulsifier based on the dispersion and fluidity of the emulsified lotion, as well as the stratification after standing. Next, the other emulsifiers were combined with the main emulsifier to determine the most effective main emulsifier/co-emulsifier combination.

#### 2.2.3. Preparation of MAD–NLCs

The high-shear ultrasound and high-pressure homogenization methods were utilized to prepare the MAD–NLCs [25]. Briefly, the solid lipids, liquid lipids, main emulsifier, and MAD were mixed in a glass vial and stirred at 500 rpm, 85 °C to obtain the oil phase. The water phase containing the co-emulsifier was preheated at the same temperature. Then, the oil phase was transferred to the water phase at 11,000 rpm, and pre-emulsion was obtained under high shear conditions for 5 min with a High Shear Dispersion Emulsifying Machine (FM200, IKA, Staufen, Germany), followed by ultrasonication for 20 min with an Ultrasonic Cell Disruption (JY96-II, Scientz, Ningbo, China). Subsequently, the nanoparticle suspension was immediately dispersed in cold water under high-shear conditions for 1 min. Overall, MAD–NLCs were acquired after an ice bath for 10 min. The pre-emulsion was circulated four times in a High-Pressure Homogenize (AH-BASIC, ATS, Suzhou, China) at 800 bar to obtain large batches of MAD–NLCs for the follow-up studies.

#### 2.2.4. Box–Behnken Response Surface Analysis

A three-factor, three-level Box–Behnken experimental design was utilized to determine the ratios of the key components and the polynomial models for optimizing the MAD–NLCs, including the ratio of solid lipids to solid–liquid mixed lipids (SL), emulsifiers to solid–liquid mixed lipids (EL), and MAD to solid–liquid mixed lipids (ML). The hydrodynamic diameter (HD) and zeta potential (ZP) were selected as the indicators, and Design-Expert 8.0 software (State-Ease, Inc., Minneapolis, MN, USA) was applied to determine the optimal formulation of MAD–NLCs. The factors and levels of the Box–Behnken design are displayed in Table 1.

### 2.3. Characterization of MAD–NLCs

#### 2.3.1. Hydrodynamic Diameter, Polydispersity Index, and Zeta Potential of MAD–NLCs

The prepared MAD–NLCs were stored overnight and diluted to an appropriate concentration to determine the hydrodynamic diameter (HD), polydispersity index (PDI), and zeta potential (ZP) by dynamic light scattering (DLS) using a Zetasizer Nano ZS90 (Malvern, PA, USA) at an angle of 90° at 25 °C.

#### 2.3.2. The Morphology of MAD–NLCs

The morphology of the MAD–NLCs was observed using a transmission electron microscope (TEM; Tecnai 12, Philips, The Netherlands). The diluted MAD–NLCs were placed on a 300-mesh copper net, negatively stained with 2% (*w*/*v*) phosphotungstic acid for 2 min, and then naturally dried at room temperature for the TEM analysis.

#### 2.3.3. X-ray Diffraction Research

SA, MAD, a physical mixture of MAD and SA, freeze-dried MAD–NLC powder, and freeze-dried blank NLC powder were researched using X-ray diffraction (XRD; D8 Advance, Bruker-AXS, Karlsruhe, Germany) to analyze the crystal structure changes in MAD during the formation of the NLCs. Further, the aforementioned samples were placed in a crucible and compacted on a glass slide. Subsequently, XRD equipped with a Cu/Kα radiation source was performed at 4°/min in the scanning range of 5°~50° under the conditions of 40 kV/40 mA.

#### 2.3.4. Fourier-Transform Infrared Spectroscopy

Fourier-transform infrared (FTIR) spectra were recorded for the SA, MAD, physical mixture of MAD and SA, freeze-dried MAD–NLC powder, and freeze-dried blank NLC powder using a FTIR spectrometer (IS5&N380, Nicolet, Waltham, MA, USA). Before conducting FTIR spectroscopy, the samples were mixed with potassium bromide (KBr) at a ratio of 1:100 and pressed into a pellet using a high-pressure hydraulic machine. The infrared scanning wave numbers were ranged from 4000 cm^−1^ to 400 cm^−1^.

### 2.4. Encapsulation Efficiency and Drug Loading of MAD–NLCs

High-performance liquid chromatography (HPLC; Waters e2695, Milford, MA, USA) was used to determine the encapsulation efficiency (EE) and drug loading (DL). MAD–NLCs were demulsified by methanol and ultrasonic treatment and then derivatized for HPLC detection. The derivation process was as follows: samples were mixed and reacted with glacial acetic acid and dansylhydrazine solution, then reconstituted with the mobile phase after drying under a nitrogen blow. Next, the content of MAD in the emulsion was determined by HPLC. Finally, the MAD–NLCs were centrifuged in a centrifugal ultrafiltration tube (MWCO: 100 kDa, Millipore, Bilrika, MA, USA), and the filtrate was derivatized to determine the contents of unencapsulated MAD in the emulsion by HPLC. Altogether, the EE and DL of the MAD–NLCs were calculated using the following formulas: $$\mathrm{EE}\;(\%)=\left(1-\frac{\mathrm{The}\;\mathrm{weight}\;\mathrm{of}\;\mathrm{unencapsulated}\;\mathrm{MAD}\;\mathrm{in}\;\mathrm{the}\;\mathrm{emulsion}}{\mathrm{The}\;\mathrm{weight}\;\mathrm{of}\;\mathrm{MAD}\;\mathrm{in}\;\mathrm{the}\;\mathrm{emulsion}}\right)\;\times \;100\%$$
$$\mathrm{DL}\;(\%)=\frac{\mathrm{The}\;\mathrm{weight}\;\mathrm{of}\;\mathrm{MAD}\;\mathrm{in}\;\mathrm{the}\;\mathrm{emulsion}}{\mathrm{Total}\;\mathrm{emulsion}\;\mathrm{weight}}\;\times \;100\%$$

### 2.5. In Vitro Release Study

The in vitro drug release of the MAD–NLCs and the active pharmaceutical ingredient of MAD (API) in PBS (pH 7.4, containing 1% sodium dodecyl sulfonate (SDS) to enhance the solubility of MAD) for 24 h were investigated. The samples were magnetically stirred in a cellulose dialysis bag (MWCO of 100 kD, 16 mm, and 0.79 mL/cm) at 37 °C and 100 rpm; some samples were withdrawn from the release medium at specific time intervals and concurrently supplemented with the same volume of fresh release medium. The drug concentrations were analyzed using HPLC after derivation.

### 2.6. Evaluation of Anticoccidial Effect 

#### 2.6.1. Experimental Design

One hundred and eighty 1-day-old healthy roosters without coccidia were purchased from a commercial hatchery (Qinglongshan Animal Breeding Farm, Nanjing, China). After feeding for 11 days with a diet free of anticoccidial drugs, the chickens were randomly divided into six groups with similar body weights, including the (1) uninfected–untreated group, (2) infected–untreated group, (3) MAD premix group (supplied with feed containing 5 mg/kg of MAD premix), (4) low-dose group (supplied with water containing 1.25 mg/kg of MAD–NLCs), (5) medium-dose group (supplied with water containing 2.5 mg/kg of MAD–NLCs), and (6) high-dose group (supplied with water containing 5 mg/kg of MAD–NLCs). On day 14, each chicken was orally inoculated with 5 × 10^4^ sporulated oocysts of *Eimeria tenella*, except for the uninfected–untreated group. The chickens’ mortality rates and clinical symptoms, including weight changes, feed consumptions, and bloody diarrhea, were observed and recorded during the experiment.

#### 2.6.2. Evaluation of Anticoccidial Effect

All the chickens were sacrificed, and the ceca were collected on day 22. The ceca of each group were scored using the method of Johnson and Reid [26] according to the severity of the cecum. The lesion scores ranged from 0, indicating a healthy cecum with no lesions, to 4, indicating extremely serious lesions and death. A histopathological examination of the ceca was performed as follows. Briefly, the cecum samples were firstly flushed and fixed in 10% neutral buffered formalin for 72 h. Then, the samples were rimmed and processed via dehydration in ascending grades of ethanol and clearing up in xylene, synthetic wax infiltration, and blocking into fragment tissue embedding media. Moreover, 5-μm-thick sections were treated via rotatory microtome and stained with Hematoxylin and Eosin (H&E) for a routine histopathological examination. The feces and cecal contents of each group were collected to calculate the oocyst values using the McMaster method [27]. In short, 2 g of feces were dispersed in 60 mL of saturated saline, the suspension was filled with a McFarland counter, and the total number of oocysts under two grids was recorded. The survival rate of each group was calculated, and the chickens were weighed to determine the relative growth rate after the experiment. The anticoccidial index (ACI) was used to evaluate the anticoccidial efficacy [28]: ACI = (relative weight gain rate + survival rate) − (lesion value + oocyst value). ACI < 120, a poor curative effect; 60 < ACI ≤ 120, a moderate curative effect; 160 < ACI ≤ 180, a good curative effect; and ACI > 180, an excellent curative effect.

### 2.7. Statistical Analysis

GraphPad Prism 8.0 (La Jolla, CA, USA) was used to analyze the data. ANOVA was utilized to evaluate the differences between the two groups. The data were shown as the mean ± standard deviation (SD). * *p* < 0.05 and ** *p* < 0.01 were considered statistically significant and extremely statistically significant.

## 3. Results and Discussion

### 3.1. Preparation of MAD–NLCs

#### 3.1.1. Screening of Lipids

Usually, the solubility of the drug in the lipids determined the selection of lipids during the preparation of NLCs, as it has a direct impact on the EE and DL of the NLCs [29]. The solubility of MAD in various solid lipids at 15 °C above their melting point is displayed in Figure 1A. Stearic acid (SA) with the highest solubility for MAD was finally selected as the solid lipid at a preparation temperature of 85 °C. SA displayed a neat and clear crystal structure under the light microscope (Appendix A), but the addition of MAD damaged this structure and caused the surface to become smooth and disordered (Appendix A), demonstrating the compatibility between SA and MAD. The solubility of MAD in the liquid lipids is illustrated in Figure 1B. The MAD did not dissolve in either castor oil or squalene, but it was highly soluble in oleic acid (OA) that was often applied as a liquid lipid in NLCs [30]. Subsequently, the solubility of MAD in different ratios of SA and OA is indicated in Figure 1C. The solubility of MAD achieved the largest value when the ratio of SA and OA was 1:9, and thus, the ratio of the mixed lipids was chosen as 1:9 in the subsequent studies.

#### 3.1.2. Screening of Emulsifiers to Mixed Lipids

Emulsifiers were incorporated into the NLC formulations for dispersing one immiscible phase into another during their fabrication [31]. Generally, the combination of two emulsifiers (a main emulsifier and a co-emulsifier) decreases the HD [32]. The effects of various emulsifiers on the solid–liquid mixed lipids is displayed in Appendix A, which illustrates the most effective main emulsifiers as Tween 20 and Tween 80. Poloxamer 188 and Poloxamer 407 were selected as co-emulsifiers based on their HLB values [33] and combined with the main emulsifier to investigate their performance. Accordingly, the group of Tween 20 and Poloxamer 188 with the best emulsification effect was chosen as the main emulsifier/co-emulsifier combination.

#### 3.1.3. Box–Behnken Response Surface Analysis

The response surface method combines mathematical and statistical techniques to facilitate the analysis and modeling of the formulation obstacles [34], process parameters, and calculate the relationship between the obtained response surfaces and the controllable input parameters [35,36]. Seventeen experimental runs with five repeated center points were required for the sake of attaining a more uniform estimation of the prediction variance over the entire design space in this model [37]. The experimental design and results created by Design-Expert software are presented in Table 2. Subsequently, the results of seventeen groups were analyzed and are presented in Appendix A, with the quadratic polynomial regression equation between the HD and three factors: HD = 348.42 + 82.93A + 22.99B + 20.96C + 35.03AB − 12.22AC − 1.70BC − 11.59A2 − 3.91B^2^ + 3.39C^2^
and the quadratic polynomial regression equation between the ZP and three factors: ZP = −30.78 + 1.39A + 1.35B + 0.56C + 1.70AB + 0.075AC − 0.30BC − 3.30A^2^ + 2.18B^2^ − 5.55C^2^. 

The results in Appendix A illustrate the differences between varied treatments of the HD model, and the ZP model was highly significant (*p* < 0.01), whereas the lack of fit was not significant (*p* > 0.05), indicating the residuals entirely resulted from random errors. The regression equation coefficient R^2^ and adjusted R^2^, which reflect the variety of the response values, were both greater than 90%, demonstrating that both prediction models were reliable.

The differences in the linear coefficients were extremely significant in the HD model, as was the interaction coefficient AB, though the other interaction coefficients were not. Accordingly, the influence order of each factor on the HD was SL > EL> ML.

The linear coefficients A and B were extremely different in the ZP model, and meanwhile, all the quadric coefficients were highly significant. Additionally, the interaction coefficient AB was significantly different, and the others were not. Altogether, the influence order of each factor on the ZP was SL > EL > ML.

Three-dimensional response surface images were generated based on the above data, as shown in Figure 2. The smaller the proportion of SA, the smaller the HD. The addition of liquid lipids decreased the viscosity and surface tension of the system, facilitating the formation of small droplets and decreasing the HD [38]. The emulsifiers can attenuate the tension at the interface between the water phase and the lipid phase to form smaller emulsion droplets, impacting the HD [31]. The MAD loaded into NLCs increased the HD of the system (Figure 2A) by increasing the drug concentrations [39]. In addition, the ZP of the NLCs increased and then decreased with the increase of the SA and MAD contents. The emulsifiers also affected the ZP, perhaps due to the coats formed on the surfaces of the NLCs (Figure 2B) [40].

The optimized MAD–NLCs formulation according to Design-Expert software was 10% SL, 30.58% EL, and 30% ML. The HD predicted by the software was 223.50 nm, and the ZP was −41.50 mV. Subsequently, the optimal formulation was verified by producing MAD–NLCs with 10% SL, 31% EL, and 30% ML. Overall, the HD and ZP of the three parallel experiments were compared with the predicted values (Appendix A), and the error percentage was calculated by the following formula. The observed values of HD and ZP for the optimized MAD–NLCs were 214.1 ± 11.90 nm and −42.8 ± 1.05 mV, respectively, which were very close to the predicted values of the equation. Thus, the optimal preparation method for the MAD–NLCs designed by the Box–Behnken response surface technique was accurate and reliable.
$$\mathrm{Error}\;(\%)=\frac{\mathrm{Observed}\;\mathrm{value}\;-\;\mathrm{Predicted}\;\mathrm{value}}{\mathrm{Predicted}\;\mathrm{value}}\;\times \;100\%$$

### 3.2. Characterization of MAD–NLCs

#### 3.2.1. The HD and ZP of MAD–NLCs

The optimized MAD–NLCs prepared by the high-pressure homogenization technique appeared as a uniform and opaque milky white liquid with excellent fluidity and superior solubility in water. Concomitantly, the MAD–NLCs had an HD of 153.6 ± 3.044 nm (Figure 3A) and a PDI of 0.174 ± 0.066, indicating the particle size distribution was relatively uniform. Notably, the larger the absolute value of the ZP, the more stable the system; a system is relatively stable when the absolute value of the ZP is greater than 30 mV [41]. The ZP of the MAD–NLCs was −41.4 ± 1.10 mV (Figure 3B), proving their excellent stability. 

#### 3.2.2. The Morphology of MAD–NLCs

The morphology of the MAD–NLCs was observed via transmission electron microscopy (TEM). As shown in Figure 3C, the MAD–NLCs particles were round or oval with a uniform size distribution and were not adhered to each other. A light ring appeared around the dark core of the NLCs, suggesting that the emulsifiers formed a coat over the surface of the NLCs to hinder the aggregation of the particles [42]. The size of the synthesized particles ranged from 10 to 70 nm with a mean diameter of 41.30 ± 14.42 nm (Figure 3D), which was less than the hydrodynamic diameter. This was because the DLS samples were highly hydrated while the TEM samples had to be dry [43,44]. 

#### 3.2.3. XRD Analysis 

The X-ray diffraction pattern of the MAD (Figure 3E) included multiple strong diffraction peaks, indicating a crystal structure. These peaks could still be observed in the physical mixture of MAD and SA, while the characteristic diffraction peaks were shifted in the MAD–NLCs, indicating a structure change of the MAD in the NLCs. Furthermore, the diffraction peaks of the blank NLCs matched the MAD–NLCs, showing that the MAD was effectively encapsulated in the NLCs and existed in an amorphous state [45]. 

#### 3.2.4. FTIR Analysis

The FTIR spectra of the SA, MAD, physical mixture of MAD and SA, freeze-dried MAD–NLC powder, and freeze-dried blank NLC powder are shown in Figure 3F. The absorption peaks of MAD at around 3222 and 1576 cm^−1^ were the stretching and deformation vibrations of the N–H bond in ammonium ion, respectively [46]. The strong absorption peak at 2954 cm^−1^ was the antisymmetric stretching vibration peak of the C–H bond in methyl and methylene, and the appearance of the peak at 1454 cm^−1^ was assigned to the deformation vibration of the methyl group [47]. The C–O–C bond (saturated cyclic ether) antisymmetric stretching vibration peak appeared at 1090 cm^−1^ [48]. Similar peaks could be witnessed in the physical mixture of MAD and lipids, whereas the characteristic absorption peaks of MAD shifted in the MAD–NLC spectra, with the N–H (ammonium ion) peak at 3392 and 1717 cm^−1^ and the C–O–C (ether) peak at 1054 cm^−1^, illustrating the changed structure of MAD. The similar spectra of the blank NLCs and MAD–NLCs demonstrated that the MAD was encapsulated into the NLCs in an amorphous state, in line with the results of the XRD analysis.

#### 3.2.5. EE and DL of MAD–NLCs

The EE and DL of the optimal MAD–NLCs measured by HPLC were 90.49 ± 1.05% and 2.34 ± 0.04%, respectively. The high EE and DL may have arisen because MAD was embedded in the lipid matrix of the NLCs in an amorphous state. It can be concluded that the NLCs loaded with MAD were successfully prepared during the experiment.

### 3.3. In Vitro Release of MAD from NLCs

The in vitro release curves of the MAD–NLCs are shown in Figure 4. The cumulative release rate of the MAD–NLCs from 0 to 12 h was only 19.93 ± 3.58%, while that of the API was 65.26 ± 0.28%, demonstrating the slow-release characteristic of the MAD–NLCs. The cumulative release rate of the MAD–NLCs in PBS for 24 h (22.64 ± 1.65%) was significantly lower than that of the API (70.47 ± 0.89%). The MAD–NLCs exhibited a slow-release behavior in PBS, which may be attributed to the almost complete encapsulation of the MAD in the NLCs.

### 3.4. Evaluation of Anticoccidial Effect of MAD–NLCs

#### 3.4.1. Clinical Symptoms and Pathological Examinations

Throughout the experiment, the uninfected–untreated group remained healthy, without death or oocysts in the feces. The infected–untreated group exhibited clinical symptoms of coccidiosis, such as a loss of appetite, dullness, bloody stools, and mass mortality of the chickens. The ceca of the dead chickens were highly swollen, dark purple in color, and full of blood. These are typical clinical symptoms and pathological features of coccidia infection [49]. The symptoms of the treatment groups were alleviated to varying degrees, indicating the success of the coccidiosis model [50].

The ceca of the chickens in each group were taken for HE staining after the experiment, and the results are shown in Figure 5. The cecum tissue of the chickens in the uninfected–untreated group appeared normal (Figure 5A). In the infected–untreated group (Figure 5B), the cecal tissues were seriously damaged due to coccidia parasitism, and the intestinal cavity contained a large number of exfoliated epithelial cells and coccidia oocysts; obviously, the pathological change of the cecum was caused by coccidia parasitism [51]. The lesions of the ceca of the treatment groups (Figure 4C–F) were relatively mild, and the cecal structure was basically normal, with a few inflammatory cell infiltrations and a small number of oocysts. These results indicated that the MAD–NLCs were therapeutically effective against cecal lesions caused by avian coccidiosis.

#### 3.4.2. Evaluation of Anticoccidial Effect

The relative weight gain and survival rate of each group are shown in Appendix A. The weight gain and survival rate of the infected–untreated group slowly increased owing to the coccidia parasitism, which was lower than those of the other groups, wherein the MAD–NLCs inhibited the weight loss and death caused by coccidiosis [2]. Chickens in the high-dose group had a lower weight gain and survival rate than the medium-dose group, possibly due to MAD poisoning [52]. The results of the cecal lesions and oocysts are exhibited in Appendix A. The infected–untreated group had serious cecal lesions and many coccidia oocysts. The inhibitory effect of the MAD–NLCs on the cecal lesions and oocysts was clearly positively related to the dose; the ceca in the high-dose group were almost free of lesions and oocysts, showing the strong anticoccidial effect of the MAD–NLCs.

Table 3 presented the anticoccidial index (ACI) of each group. Compared with the infected–untreated group, the ACI of the treatment groups was significantly increased. The MAD–NLCs had a stronger anticoccidial effect than the MAD premix. The decreased ACI in the high-dose group may be due to MAD poisoning [53]. The chickens in the medium-dose group showed no symptoms of MAD poisoning, achieving an ACI of more than 180, which indicated a strong anticoccidial effect. Taken together, 2.5 mg/kg of MAD–NLCs dissolved in water were recommended for the effective prevention and treatment of avian coccidiosis.

## 4. Conclusions

As a drug with an excellent anticoccidial effect, MAD has a narrow therapeutic window and low water solubility, which considerably limit its clinical use. Therefore, it is urgent to develop a new formulation of MAD with an increased solubility and enhanced pharmacological effect. We developed and optimized a novel MAD formulation: MAD–NLCs. In this study, the Box–Behnken response surface method and high-shear ultrasound were applied to optimize the MAD–NLCs. The optimal MAD–NLCs formulation prepared by the high-pressure homogenization method had a high solubility, a HD of 153.6 ± 3.044 nm, and a favorable EE of 90.49 ± 1.05%, as well as slow-release characteristics. MAD is encapsulated in NLCs in an amorphous state. MAD–NLCs are expected to develop into a new preparation of MAD with high solubility and favorable characteristics. Meantime, the optimal formulation of the MAD–NLCs had a powerful anticoccidial effect. Overall, MAD–NLCs have a promising marketability, a high application value, and the potential to develop into a new formulation of anticoccidial drugs that can be administered in drinking water.

## Figures and Tables

**Figure 1 pharmaceutics-14-01330-f001:**
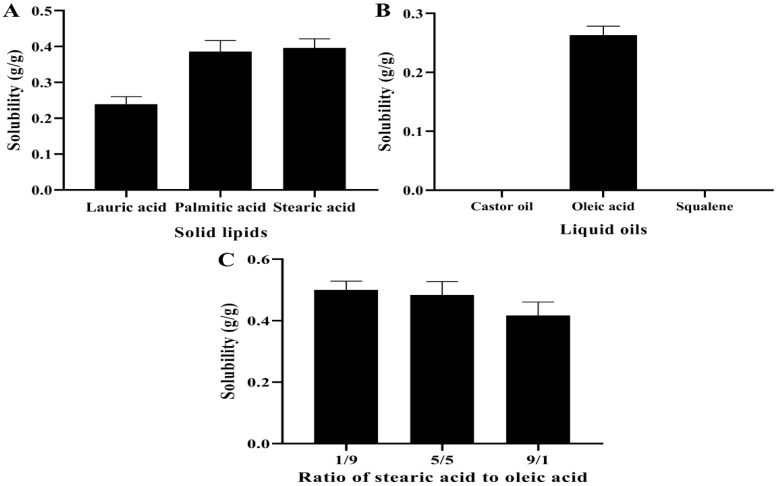
Screening of the solid and liquid lipids for the solubility of MAD. (**A**) Solubility of MAD was determined in different solid lipids. (**B**) Solubility of MAD was analyzed in various liquid lipids. (**C**) Solubility of MAD was detected in varied ratios of stearic acid and oleic acid.

**Figure 2 pharmaceutics-14-01330-f002:**
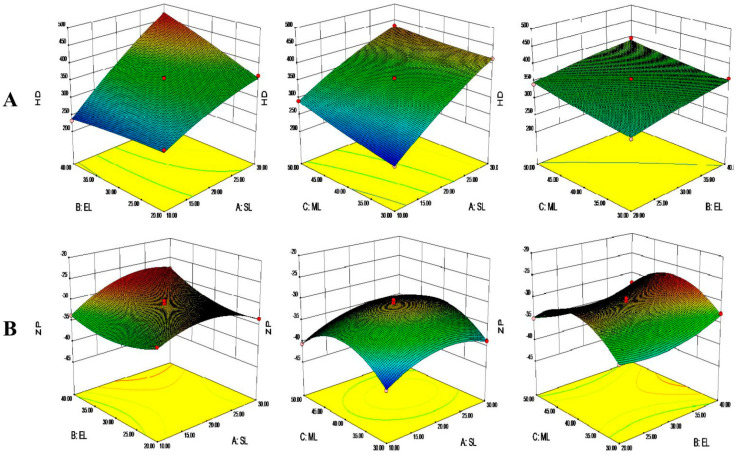
Three-dimensional response surface images of the solid lipid to lipid ratio (SL), emulsifier to lipid ratio (EL), and MAD to lipid ratio (ML) on (**A**) the hydrodynamic diameter (HD) and (**B**) zeta potential (ZP).

**Figure 3 pharmaceutics-14-01330-f003:**
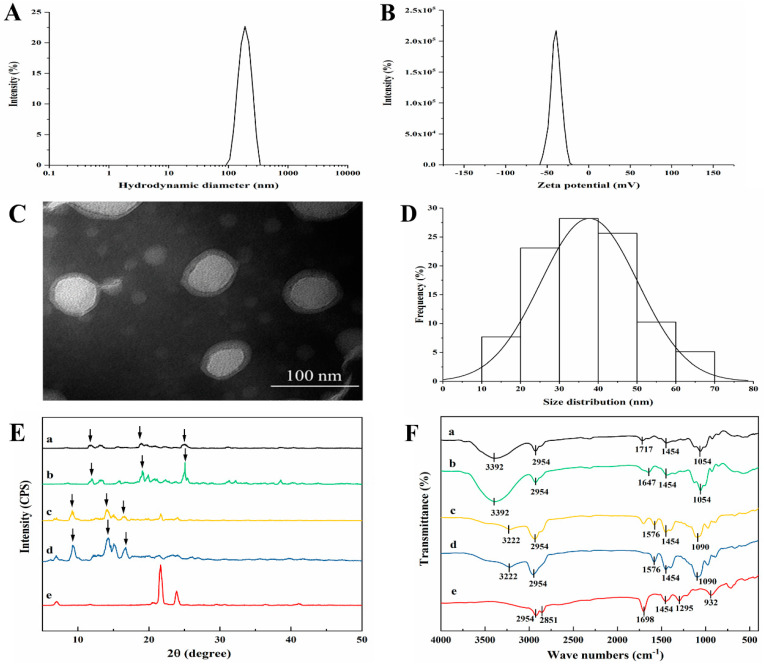
Characterization of the optimized MAD–NLCs. (**A**) Hydrodynamic diameter (HD) and (**B**) zeta potential (ZP) of the MAD–NLCs were determined by DLS. (**C**) The morphology of the MAD–NLCs was observed by TEM, and (**D**) the size distribution was obtained via analysis of the particles from several TEM images. (**E**) XRD patterns and (**F**) FTIR spectra for (a) MAD–NLCs, (b) blank NLCs, (c) the physical mixture, (d) MAD, and (e) SA were shown.

**Figure 4 pharmaceutics-14-01330-f004:**
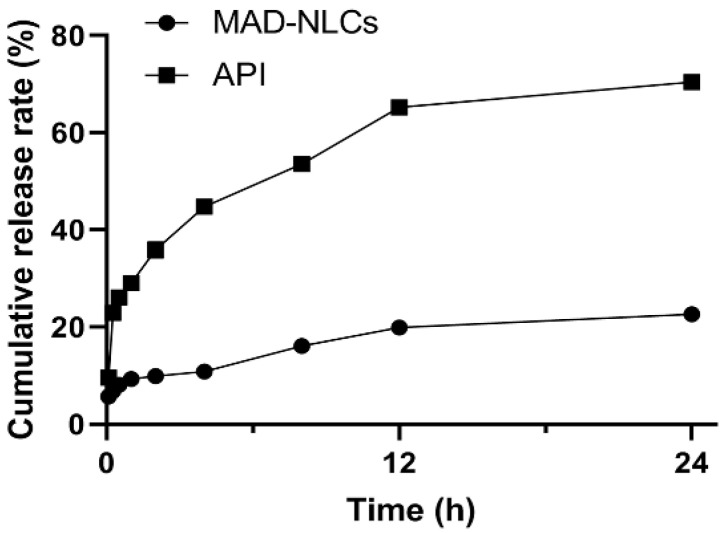
Cumulative release rates of the MAD–NLCs and MAD in PBS solution.

**Figure 5 pharmaceutics-14-01330-f005:**
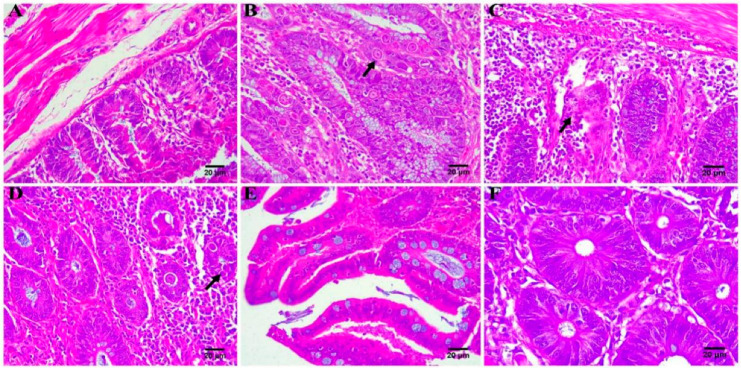
Histopathological analysis of chicken ceca (*n* = 3). (**A**) Uninfected–untreated group. (**B**) Infected–untreated group (black solid arrow represents coccidia oocysts). (**C**) MAD premix group(black solid arrow represents structural disorders). (**D**) Low-dose group of MAD–NLCs (black solid arrow represents coccidia oocysts). (**E**) Medium-dose group of MAD–NLCs. (**F**) High-dose group of MAD–NLC structural disorders.

**Table 1 pharmaceutics-14-01330-t001:** Variables in the Box–Behnken design for the formulation development of MAD–NLCs.

Independent Variables (Factors)	Levels		
	−1	0	1
A: SL (%)	10	20	30
B: EL (%)	20	30	40
C: ML (%)	30	40	50
Dependent variables (responses)	Unit	Goal	
Y1: HD	nm	Minimize	
Y2: ZP	mV	Minimize	

Note: SL: The ratio of solid lipids to the solid–liquid mixed lipids. EL: The ratio of emulsifiers to the solid–liquid mixed lipids. ML: The ratio of MAD to the solid–liquid mixed lipids. HD: Hydrodynamic diameter. ZP: Zeta potential.

**Table 2 pharmaceutics-14-01330-t002:** Experimental design and values of the responses for the MAD–NLCs formulations.

Formulation Codes	Independent Variables			Dependent Variables	
	A	B	C	Y1	Y2
1	20	30	40	358.1	−29.6
2	30	30	30	413.7	−38.8
3	20	30	40	340.2	−30.4
4	30	30	50	432.5	−37.4
5	10	30	50	291.2	−40.6
6	20	30	40	358.5	−31.5
7	30	40	40	467.6	−27.8
8	10	20	40	268.3	−32.6
9	10	40	40	231.6	−33.7
10	20	20	50	340.6	−34.9
11	10	30	30	223.5	−41.7
12	20	40	30	358.6	−32.8
13	20	30	40	328.2	−32.7
14	20	40	50	395.8	−32.4
15	20	30	40	357.1	−29.7
16	30	20	40	364.2	−33.5
17	20	20	30	296.6	−36.5

Note: A: The ratio of solid lipids to the solid–liquid mixed lipids. B: The ratio of emulsifiers to the solid–liquid mixed lipids. C: The ratio of MAD to the solid–liquid mixed lipids. Y1: Hydrodynamic diameter. Y2: Zeta potential.

**Table 3 pharmaceutics-14-01330-t003:** Evaluation results of the anticoccidial index in each experimental group.

Group	Survival Rate (%)	Relative Weight Gain (%)	Cecal Lesion	Oocyst Value	Anticoccidial Index
**1**	100	100	0	0	200
**2**	66.67	59.07	34	40	51.74
**3**	90	82.33	10.3	10	151.99
**4**	100	84.95	8.7	10	166.29
**5**	100	93.72	4	1	187.78
**6**	90	85.85	1.3	0	174.52

## Data Availability

Data are contained within the article or the Appendix A.

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
