# Peer review of "Optimization of Maduramicin Ammonium-Loaded Nanostructured Lipid Carriers Using Box–Behnken Design for Enhanced Anticoccidial Effect against Eimeria tenella in Broiler Chickens"

_pharmaceutics, 2022, doi:10.3390/pharmaceutics14071330_

Round 1
Reviewer 1 Report
The authors present the development of Maduramicin Ammonium (MAD)-loaded nanostructural lipid carriers (MAD-NLCs) optimized through a three level three factor Box-Behnken response surface design wherein the key component ratios were determined. Different parameters such as hydrodynamic diameter, zeta potential, crystal structure, FTIR profiles, in vitro drug release datas were investigated along with the anticoccidial effect against Eimeria tenella in broiler chickens. The authors conclude that MAD-NLCs posess the ability to confer augmented anticoccidial therapeutic benefits in broiler chicken when developed as a drinking preparation.
The overall context of the article is indeed interesting and particularly beneficial considering its relevance in veterinary medicine. However the manuscript has not been presented very well and the observations have not been adequately justified. Many experimental aspects are not explained scientifically and many statements lack scientific rationale. The manuscript needs to be significantly improved in this regard. The key concerns are pointed out below:
1. Authors could include few additional statements in the introductory statement describing the disease condition of coccidiosis in broiler chicken and some currently existing anticoccidial therapeutics before describing the specifics relating to MAD and need to develop alternative formulations.
2. Are there any approved NLCs in the market? Please provide few specific examples of the therapeutic utility of NLCs (with regard to some insoluble drugs)?
3. What are the specific advantages and disadvanatges of high shear ultrasound method and high pressure homogenization method for fabricating NLCs?
4. Section 2.2.1, methods : what was the rationale behind choosing the mixing ratios such as 1:9, 5:5 and 9:1?
5. How was emulsification effect assessed? (section 2.2.2)
6. Section 2.2.3: what was the stirring speed used to obtain the oil phase?
7. Section 2.2.3: is there any specific unit or figure which can be used to describe ‘high shear’ quantitatively? How large were the batches of MAD-NLCs produced?
8. What was the rationale behind adding 1% SDS to PBS for in vitro drug release study (section 2.5)?
9. Section 2.6.1- ‘the chicken’s mental state, mortality, weight changes and bloody stools were monitored every day during the whole experiment’- this does not seem to be scientifically sound experimental strategy. Please add more details and explain clearly. How was ‘the mental state’ monitored?
10. Please include the details of histological analyses in section 2.6.2.
11. ‘SA presented with a neat and clear crystal stricture under the light microscope’ – figure S1- the interpretation cannot be justified from this figure unless justified with a control image.
12. Page 7- lines 274-283. Please add references to justify the assumptions indicated in every statement. Was viscosity and surface tension measured (line 275)?
13. Section 3.2.1- line 307- ‘notably the larger the absolute value of ZP, the more stable the system will be’ – this statement needs better clarity.
14. Lines 319-320, section 3.2.2: ‘remarkably the DLS samples were highly hydrated while the TEM samples were required to keep dry’- please rewrite this statement with better clarity and explain this in the context of findings.
15. Characteristic diffraction peaks shown in XRD data (Figure 3E) need to be marked and also references should be cited in section 3.2.3.
16. Please provide references for the absorption peaks denoting specific functional groups – section 3.2.4
17. Section 3.3-line 356- is it ‘accumulative’ or ‘cumulative release rate’? What is the percentage increase between the release rate% at time points 12 vs 24hours for MAD-NLCs and APIs? Was it considerably different? The figure 4 might be confusing in this regard so please explain this in order to justify the statements given in section 3.3.
18. Lines 365-367- section 3.4.1- as previously mentioned, please rewrite the findings in a more scientific manner with supporting references.
19. Figure 5- what was the ‘n’ for each group for histological analysis? How many fields per section were imaged? Were the observations consistent and has it been pathologically confirmed? Please indicate these details.
20. What were the statistical tests used to analyze the datas presented in figures S3-S6?
21. Section 3.4.2- MAD poisoning is important considering the context of this research work and hence the main findings need to be explained in much more detailed manner. Are there any observations which possibly confirms the minimal toxicity of NLCs? These need to be highlighted.
22. The novelty of the current work relative to previous studies investigating similar causes need to be explained well in both introductory and conclusion sections.
Reviewer 2 Report
The work performed by authors on the topic “Optimization of Maduramicin Ammonium Loaded Nanostructured Lipid Carriers Using Box-Behnken Design for Enhanced Anticoccidial Effect Against Eimeria tenella in Broiler Chickens” is an interest to the reader and within the scope of the journal. But manuscript needs revision.
1. Provide the motivation behind using nanostructured lipid carriers for the synthesis of MAD-loaded nanostructured lipid carriers (MAD-NLCs) to deal with the defects of MAD.
2. Discuss several loaded carriers used for drug delivery in the introduction section.
3. Introduction section looks weak with an insufficient systematic literature survey. The introduction section needs to modify further with a more brief discussion based on literature.
4. Section 2.2.4. Box-Behnken response surface analysis, why this section paragraph is written in bold, needs to be corrected.
5. Results and discussion section need to be more elaborated and discussed in detail with proper justification.
6. Section 3.2.4. FT-IR analysis, the author has mentioned “The absorption peaks of pure MAD at around 3222 cm−1 and 1576 cm−1 were the stretching vibration and deformation vibration of N-H on the ammonium ion, respectively”. Kindly provide the reference to support the above sentence.
7. Section 4 conclusion, need to be more elaborated.
Round 2
Reviewer 1 Report
Authors have taken considerable efforts to improve the quality of the manuscript from the previous version which is reflected in the current version. Relevant discussion statements explaining the results and literature references supporting the statements have been added to all important sections. All questions raised in the first review have been answered. However, there are very minor concerns with few answers which can be easily addressed by the authors. The manuscript can thus be considered for publication.
1. Section 3.3- "the 24h cumulative release rate of the MAD from the NLCs was always within 30%, while that of the API reached almost 80% at 24h"- this sentence can be omitted as it does not provide the exact figures; release rates need to be scientifically represented as exact figure with standard deviation. However, this has been indicated in the next sentence, therefore the above line is not needed.
2. ANOVA statistical test details need to be included in methods section as the last section describing statistical analyses. The figures representing the analyzed data should also contain the details in figure labels.
3. Histological data in figure 5. Label needs details such as 'n' value and sections analyzed.
Reviewer 2 Report
The authors have satisfactorily revised the manuscript.
Author Response
Dear reviewer,
Thank you for your encouragement and recognition of our revised manuscript entitled “Optimization of Maduramicin Ammonium-Loaded Nanostructured Lipid Carriers Using Box-Behnken Design for Enhanced Anticoccidial Effect against Eimeria tenella in Broiler Chickens”. Your approval made all of us buck up. Thanks again for your patience and effort.
Sincerely yours,
Dawei Guo
